# Avoiding mode collapse in diffusion models fine-tuned with reinforcement learning

## Abstract

Fine-tuning foundation models via reinforcement learning (RL) has proven promising for aligning to downstream objectives. In the case of diffusion models (DMs), though RL training improves alignment from early timesteps, critical issues such as training instability and mode collapse arise. We address these drawbacks by exploiting the hierarchical nature of DMs: we train them dynamically at each epoch with a tailored RL method, allowing for continual evaluation and step-by-step refinement of the model performance (or alignment). Furthermore, we find that not every denoising step needs to be fine-tuned to align DMs to downstream tasks. Consequently, in addition to clipping, we regularise model parameters at distinct learning phases via a sliding-window approach. Our approach, termed Hierarchical Reward Fine-tuning (HRF), is validated on the Denoising Diffusion Policy Optimisation method, where we show that models trained with HRF achieve better preservation of diversity in downstream tasks, thus enhancing the fine-tuning robustness and at uncompromising mean rewards.

## 1 Introduction

Diffusion models (DMs) are the *de facto* state of the art in prompt-based generative modelling across various tasks including text-to-image, text-to-video, molecular graph modelling and medical image reconstruction (Ramesh et al., 2021; Rombach et al., 2022; Ho et al., 2022; Singer et al., 2023; Jing et al., 2022; Song et al., 2022). Most of these applications build on the original Denoising Diffusion Probabilistic Model (DDPM) by Ho et al. (2020), but the extension to other formulations and variants is fast growing. It includes Score-based and Flow matching generative models, among others (Song and Ermon, 2019; Song et al., 2020; Lipman et al., 2022).

Recent works use Reinforcement Learning (RL) to align DMs to downstream tasks that are otherwise difficult to address, e.g., via explicit class labelling, such as generating *aesthetic* images following a specific prompt, or producing images admitting significant JPEG compression rates. These methods, such as the Denoising Diffusion Policy Optimization (DDPO) (Black et al., 2024) and others (Deng et al., 2024; Fan et al., 2023), formulate the denoising steps as a Markov Decision Process (MDP) controlled through a reward function. Despite recent works (Ouyang et al., 2022; Schulman et al., 2017) making RL fine-tuning more stable and accurate, the consequence for alignment in reward-based DDPM extensions are skewed generated samples, which lead to *mode collapse*, i.e., when the model's inherent diversity vanishes. Fig. 1 shows samples from DDPM and DDPO to illustrate that, although DDPO provides better samples than DDPM (in terms of the LAION aesthetic score in this case), it sacrifices sample diversity: the images look the same.

We build on the hierarchical interpretation of DDPMs (Sclocchi et al., 2024) and propose a methodology called *Hierarchical Reward Fine-tuning* (HRF), which performs reward-based learning from different timesteps in the diffusion using a sliding window approach. In DDPM, the HRF window selection mechanism fixes high-level features and generates variations in low-level features, facilitating exploration while preserving the sample's semantics, as illustrated in Fig. 3. To achieve this, we identify distinct stages in the learning process, distinguishing between high- and low-level features, and apply online RL at each stage, thus effectively training intermediate steps in the diffusion. Our proposal thus enables a controlled learning scheme promoting sample diversity, mitigating mode collapse, and thus successfully aligning diffusion models (DMs) to downstream tasks.

The proposed HRF is implemented and experimentally assessed in the generation of RL-optimised diversity-preserving samples within DDPM. To this end, we consider a DM pre-trained on CelebaHQ (Karras et al., 2018), and fine-tuned over three downstream tasks originally considered in (Black et al., 2024): compressibility, incompressibility and LAION aesthetic score. Fig. 4 shows a succinct illustration of the results of our method for each of these tasks.

The key contributions of our work are:

- Insights into the learning dynamics of DDPMs when fine-tuned via online RL.
- A critical assessment of the state of the art, in terms of its inability to preserve sample diversity due to an overoptimisation in noisy (early) stages of the diffusion chain.
- A novel framework for preserving sample diversity in DDPMs fine-tuned with RL, based on a hierarchical interpretation of diffusion models, called Hierarchical Reward Fine-tuning (HRF).
- An experimental validation of HRF demonstrating: i) advantages in controlled learning, ii) reduced need for reward handcrafting, and iii) improved preservation of sample diversity compared to DDPO.

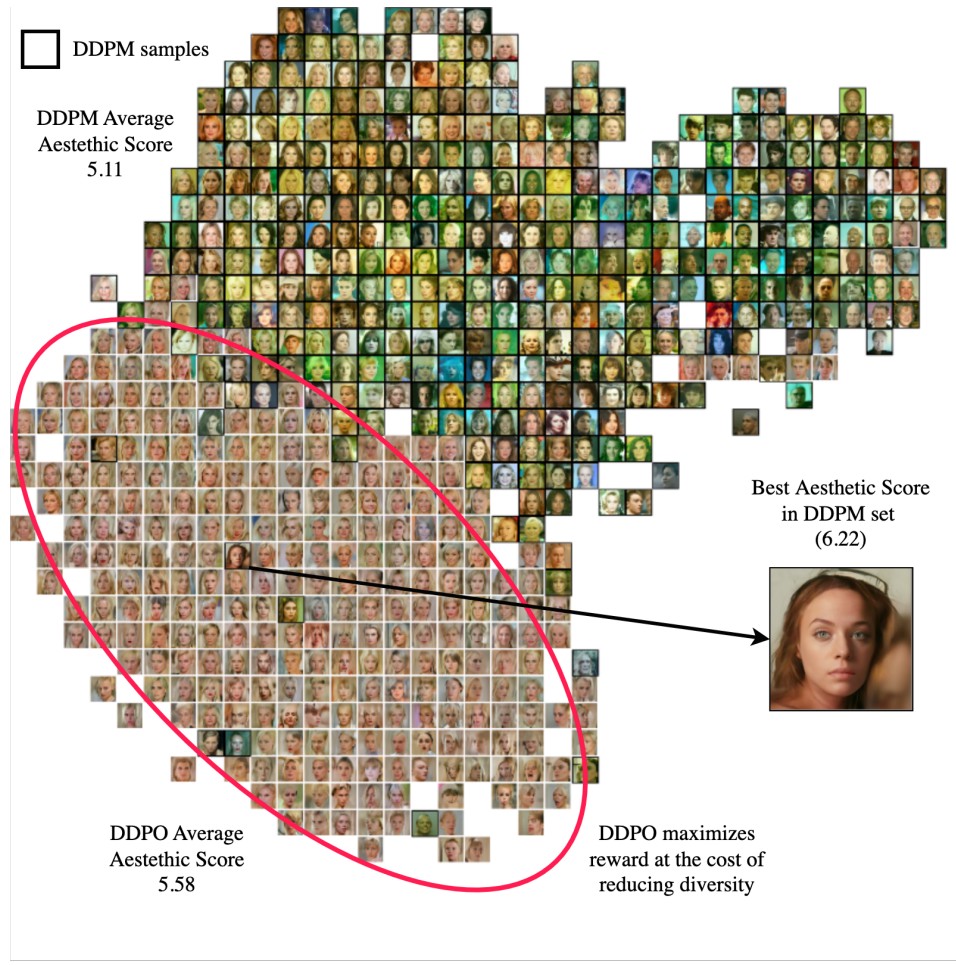

Figure 1: **Comparison of Image Synthesis Using CelebA-HQ-Based Models.** 2D projection of CLIP embeddings for two sets of 1,000 samples: i) DDPM samples (black borders) and ii) DDPO samples fine-tuned with the LAION aesthetic reward (white borders). The DDPO samples were optimized to achieve a higher average aesthetic score (5.58 vs. 5.11), indicating better *aesthetic quality*. Notably, the DDPO samples cluster more tightly (red ellipse) around the highest-scoring DDPM sample, indicating a ***mode collapse effect***. Both sets of samples were generated using the same seed.

## 2  RELATED WORK

### 2.1  HIERARCHICAL FEATURES OF DIFFUSION MODELS

Recent studies have highlighted the hierarchical nature of data generation in diffusion models, particularly in the case of images. For instance, Sclocchi et al. (2024) found that a phase transition occurs during the backward diffusion at a specific timestamp. Beyond this point, the probability of reconstructing high-level features drops rapidly, while low-level features (details) change slower. High-level features typically refer to global attributes, such as overall face structure or hair type, whereas low-level features involve finer details, like skin texture or small facial details. This implies that high-level features are more susceptible to temporal changes in the diffusion process, whereas low-level features remain relatively stable across the diffusion. This observation is pivotal in our work, as we will develop a methodological approach to learn favourable representations for downstream tasks at distinct steps in the diffusion process. The main rationale behind our diversity-preserving approach is to target the early steps of the diffusion process trajectory we are training on.

### 2.2  DIFFUSION MODEL TRAINING AT DIFFERENT NOISE LEVELS

The learning dynamics and convergence properties of DDPM at different stages of the diffusion process have been studied by Hang et al. (2023), finding that convergence speed is related to the learning difficulty associated with each timestep. Additionally, Xu et al. (2024) adopted a perspective of curriculum learning and found that DDPMs learn the early steps easier than the later ones across the denoising procedure. This curriculum learning approach allows for the design of learning strategies tailored for each step of the diffusion, thus improving the convergence speed.

### 2.3  TRAINING DIFFUSION MODELS WITH REINFORCEMENT LEARNING

Approaches to train DDPMs by modelling the diffusion chain as an MDP leverage different loss formulations and regularization techniques (Black et al., 2024; Fan et al., 2023; Deng et al., 2024). However, they all build upon the same RL setup: they refine loss definitions and mitigate the degeneration of sample diversity through regularization and hyperparameter tuning, but they do not exploit the hierarchical nature of data generation inherent in diffusion models. This means that although these approaches successfully fine-tune the models for downstream tasks in terms of their achieved rewards, they do so struggling to propagate learning across all diffusion steps in a controlled manner, ultimately compromising diversity. Our contribution aims to address this limitation.

## 3  BACKGROUND

### 3.1  DIFFUSION MODELS

Let us consider denoising diffusion probabilistic models (DDPMs) (Ho et al., 2020; Sohl-Dickstein et al., 2015), which represent a distribution $p(x_0|c)$ over data samples $x_0 \in \mathcal{X}$ conditioned on contexts $c \in \mathcal{C}$. This distribution is defined by reversing the forward Markovian process $q(x_t|x_{t-1}, c)$, where the chain $\{x_t\}_{t=0}^T$ is a sequence of samples with increasing levels of noise such that $x_0$ represents a clean data sample and $x_T$ a completely noisy one, where all the data structure has been broken.

The transition probability of the backward process is typically modelled as a Gaussian with a learnable mean $\mu_\theta(x_t, c, t)$ and a fixed variance $\sigma_t^2$ (Ho et al., 2020; Song et al., 2021), that is,

$$p_\theta(x_{t-1}|x_t, c) = \mathcal{N}(x_{t-1}|\mu_\theta(x_t, c, t), \sigma_t^2 I). \tag{1}$$

The mean $\mu_\theta(x_t, c, t)$ is usually parameterised by a neural network trained with the objective:

$$L_{\text{DDPM}}(\theta) = \mathbb{E}_{(x_0,c)\sim p(x_0,c), t\sim U\{0,T\}, x_t\sim q(x_t|x_0)} \left[ \|\mu(x_0, t) - \mu_\theta(x_t, c, t)\|^2 \right], \tag{2}$$

where $\mu(x_0, t) = \mathbb{E}(x_t|x_0)$ is the expectation of $x_t$ under the forward process defined by the transition kernel $q(x_t|x_{t-1})$. This objective maximizes a variational lower bound on the log-likelihood of the data (Ho et al., 2020; Luo, 2022).

Once the DDPM is trained, i.e., the neural net $\mu_\theta(x_t, c, t)$ is learned, data generation occurs via sampling. This starts by first drawing a pure-noise sample $x_T \sim \mathcal{N}(0, I)$ and then realising the

reverse process in equation 1 to produce the sequence (or trajectory) $\{x_T, x_{T-1}, \ldots, x_0\}$, where $x_0$ is the desired sample.

## 3.2 Diffusion model as a sequential decision-making process

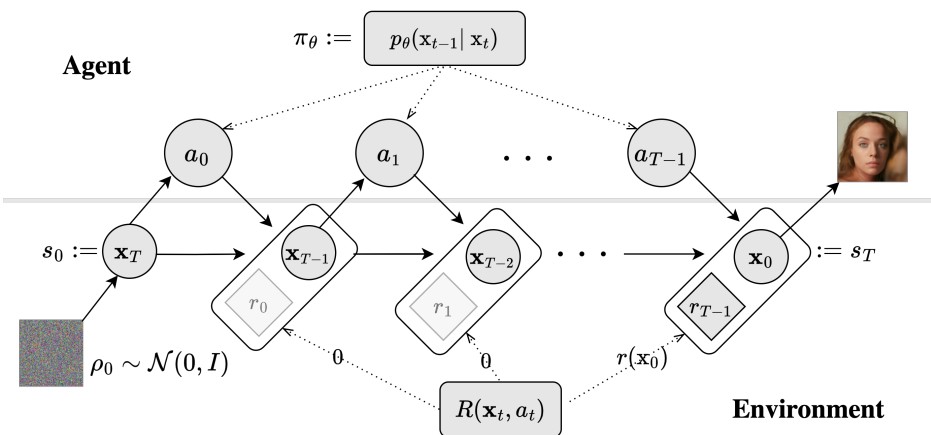

Figure 2: **Equivalence of the backward process of a diffusion model as a sequential decision-making process.** The initial state distribution of this MDP corresponds to an isotropic Gaussian, $\rho_0(s_0) \sim \mathcal{N}(0, I)$, where we assign the noise instance to the initial state $s_0 = x_T$. The agent follows a sequence of decisions $a_t$ determined by the policy $\pi_\theta(a_t \mid s_t) := p_\theta(x_{T-t-1} \mid x_{T-t})$, moving from a noisy state $x_{T-t}$ to a less noisy one $x_{T-t-1}$ until it reaches to the sample $x_0$, illustrated as the terminal state $s_T$ in the diagram. This process generates the whole denoising trajectory $\tau = \{x_T, x_{T-1}, x_{T-2}, \ldots, x_0\}$, which is associated with a reward. In the case of DDPO, the reward model $R$ only depends on the final, i.e., sample $r(x_0)$.

As a starting point, let us consider the denoising diffusion policy optimization (DDPO) formulation (Black et al., 2024). In this setting, the DDPM backward process can be interpreted as an MDP, where the policy describes how an agent moves from a state $s_t$ with a noisy sample $x_{T-t}$ to a state $s_{t-1}$ with a cleaner sample $x_{T-t-1}$, through its *denoising actions* $a_t : x_{T-t} \to x_{T-t-1}$. This occurs from the initial state $s_0$, where a noise sample is drawn from an isotropic Gaussian distribution $\rho_0 \sim \mathcal{N}(0, I)$ and assigned to $x_T$, until arriving at a terminal state $s_T$, where the sample $x_0$ is generated. Fig. 2 illustrates this RL formulation of DDPM.

In DDPO, a *denoising* neural network $p_\theta$ is used to directly estimate $s_t$ at timestep $t-1$ (or, indirectly, estimate the corresponding noise), defining the policy $\pi_\theta(a_t, s_t)$. In this post-training framework, the diffusion model parameters $\theta$ can be directly optimized via policy gradient estimation to maximize any arbitrary scalar-reward signal over the sample. $x_0$ In other words, the agent learns to denoise trajectories in order to maximize the expected reward, leveraging the denoising diffusion reinforcement learning (DDRL) objective (Black et al., 2024):

$$\mathcal{J}_{\text{DDRL}}(\theta) = \mathbb{E}_{c \sim p(c), x_0 \sim p_\theta(x_0 | c)}[r(x_0, c)]. \tag{3}$$

A critical aspect of this formulation is that the reward $R(s_t, a_t)$ **only considers the final sample** $x_0$, neglecting every non-terminal state $s_T$, or equivalently samples $x_{t \neq 0}$, as depicted in Figure 2.

Furthermore, DDPOs compute gradients either via a score function method, also known as RE-INFORCE (Schulman et al., 2015), or optimising a surrogate objective via importance sampling (Schulman et al., 2017). The latter, denoted DDPO$_{\text{IS}}$ with gradient given by

$$(\text{DDPO}_{\text{IS}}) \ \nabla_\theta \mathcal{J} = \mathbb{E}_{x_{T:0} \sim p_{\theta_{\text{old}}}} \left[ \sum_{t=0}^{T} \frac{p_\theta(x_{t-1}|x_t)}{p_{\theta_{\text{old}}}(x_{t-1}|x_t)} \nabla_\theta \log p_\theta(x_{t-1}|x_t) r(x_0) \right], \tag{4}$$

is used as a baseline in our paper to compare the proposed method.[1] Note that computing $p_\theta$ and $\log p_\theta$ in equation 4 is straightforward when $p_\theta$ is a conditional Gaussian distribution.

---

[1] From now on, when we refer to DDPO, we mean DDPO$_{\text{IS}}$ unless specified otherwise.

### 3.3 DIVERSITY LOSS DUE TO OVERPARAMETERIZATION IN THE EARLY PHASES OF DENOISING

As discussed in Secs. 2.1 and 2.2, learning in DDPM models is more effective during the noisy stages of diffusion and becomes increasingly difficult as noise is reduced. We confirm that this trend also applies to RL-based diffusion via a succinct experimental example. By employing injection sampling, a process where sampling is started with the fine-tuned model and then switched to the base model at step $t$ to continue the sampling, we observed that interventions made in the noisier stages of the denoising process have the most significant impact on the resulting samples. A detailed description of the experiment can be found in Appendix A.1.

## 4 HIERARCHICAL REWARD FINE-TUNING FOR DIFFUSION MODELS

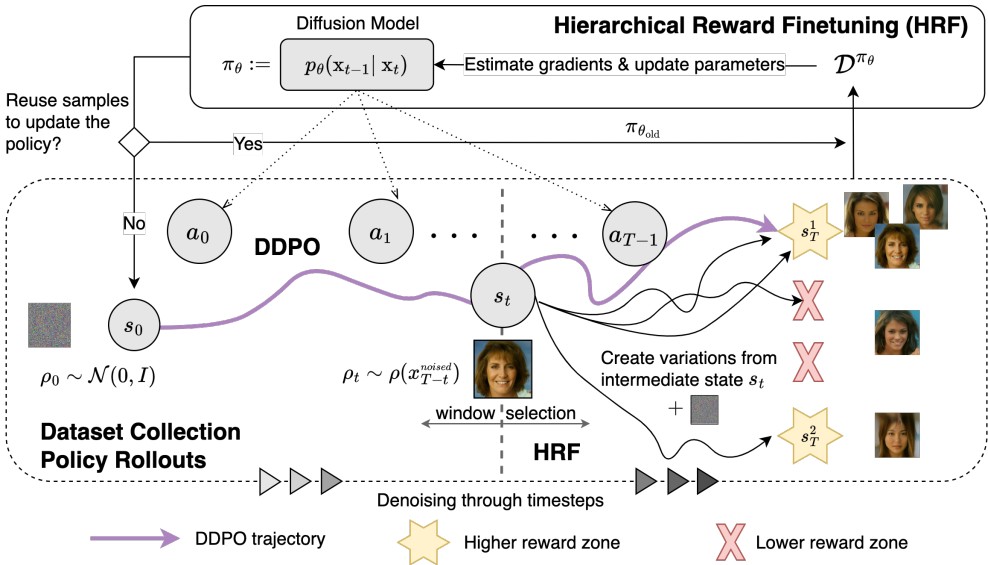

Figure 3: **Hierarchical Reward Fine-tuning (HRF)**. In **DDPO** (purple), the entire denoising trajectory is influenced, affecting both high- and low-level features. In contrast, **HRF** intervenes at later timesteps of the trajectory, selecting an specific timestep $t$ based on a *window selection schema*. At this point, an intermediate state $s_t \sim \rho_t$ is drawn from a new prior distribution, primarily adjusting low-level features while preserving high-level features and diversity, yet still achieving high rewards ($s_T^1$ yellow star). During policy rollouts, **HRF** generates divergent trajectories from $s_t$ by introducing noise at intermediate timesteps. This serves as a *low-level feature exploration mechanism* to discover regions with higher reward potential ($s_T^2$ yellow star) given the high-level information set by the new prior $\rho_t$. In both cases, the dataset of trajectories and rewards $\mathcal{D}^{\pi_\theta}$ is used to estimate the gradients via Monte Carlo sampling, which are then applied to update the diffusion model parameters. **HRF-D** dynamically adjusts the vertical window selection line during fine-tuning.

This section describes our main contribution by defining the problem statement and the methodology employed to solve it.

### 4.1 HIERARCHICAL REPRESENTATION OF TRAINING

Our approach, HRF, is underpinned by the hierarchical nature of sample generation in diffusion models as identified by Sclocchi et al. (2024). With this in mind, we propose the training diagram illustrated in Figure 3. Each training step starts at a noisy step $t$ using an image prior, guiding the model to learn from favorable policies at that stage. This is achieved by computing the trajectories and their corresponding rewards for every chosen initial step and then propagating the learning to those starting points. This approach leverages the fact that structural changes in generated samples occur at different stages of the diffusion chain, where the rate of change varies depending on the learning difficulty (Xu et al., 2024; Hang et al., 2023). The methodology remains flexible enough to

allow evaluation on fully denoised samples for each step $t$, while also facilitating the assessment of reward functions at intermediate steps. This capability helps in selecting better policies throughout the diffusion chain.

We propose two variants of the aforementioned methodology based on the window selection mechanism: HRF, which uses manually defined windows for selecting initial steps, and HRF-D, which employs a *dynamic window sampling* strategy. Refer to Section 4.3 for more details about the window selection schemes.

A major benefit of our proposed methodology is the control over when and where to optimize the DMs distinct phases, particularly for training using RL on downstream rewards that are hard to describe explicitly. Furthermore, HRF is independent of the chosen loss definitions since it still uses the final sample to represent the information conveyed in the segment of the denoising trajectory under evaluation. Another significant benefit of HRF comes from splitting the learning task into multiple training steps (abiding by the hierarchical nature of training), which allows us to optimize hyper-parameters or skip phases in a controlled manner.

## 4.2 Trajectory of Interest Sampling

We define each denoising step in the diffusion chain as $p_\theta(\mathrm{x}_{t-1}|\mathrm{x}_t, c)$. For a given time step $t$, we use importance sampling so that the objective function defined in DDPO with our modified trajectory window is:

$$(\mathrm{DDPO_{IS_{window}}}) \ \nabla_\theta \mathcal{J} = \mathbb{E}_{\mathrm{x}_{0:t} \sim p_{\theta_{old}}} \left[ \sum_{t=0}^{t} \frac{p_\theta(\mathrm{x}_{t-1}|\mathrm{x}_t, c)}{p_{\theta_{old}}(\mathrm{x}_{t-1}|\mathrm{x}_t, c)} \nabla_\theta \log p_\theta(\mathrm{x}_{t-1}|\mathrm{x}_t, c) r(\mathrm{x}_0) \right], \quad (5)$$

where the likelihood is computed over the modified diffusion trajectory. This is from $t \to 0$, as its computed over the intermediate step selected by the window selection mechanism to the final state. We also employ trust regions to address inaccurate estimations of $p_\theta$.

To maintain the consistency of our MDP definition, and as importance sampling operates over the batch where the likelihood is computed, it is important to ensure that the initial sampling step originates from the same distribution. In the original formulation of DDPO, this is achieved by defining the initial state as a sample from random noise. To replicate these conditions for each batch, we start with a clean image, add random noise up to step $t$ a total of $n$ times, based on a scheduler, and generate trajectories for $n$ samples per batch.

Our MDP definition could be written as:

$$s_t = (c, t, \mathrm{x}_t) \quad \pi(a_t \mid s_t) = p_\theta(\mathrm{x}_{t-1} \mid \mathrm{x}_t, c) \quad P(s_{t+1} \mid s_t, a_t) = (\delta_c, \delta_{t-1}, \delta_{\mathrm{x}_{t-1}})$$

$$a_t = \mathrm{x}_{t-1} \quad \rho_0(s_0) = (p(c), \delta_t, \rho(x_{\mathrm{noised}})) \quad R(s_t, a_t) = \begin{cases} r(\mathrm{x}_0, c) & \text{if } t = 0 \\ 0 & \text{otherwise} \end{cases}$$

where $\rho(x_{\mathrm{noised}})$ is the distribution of the initial state at step $t$, which is obtained by adding noise sampled from $\mathcal{N}(0, I)$ to a clean image $\mathrm{x}_0$ up to step $t$.

For $t = T$, the distribution of the initial state is equal to $\mathcal{N}(0, I)$, which is the original definition provided by DDPO. Generally, this framework is flexible enough to be adapted to any method that uses the original MDP formulation for diffusion models.

## 4.3 Window Selection

We evaluate two window selection schemes: (1) predefined windows that determine initial sampling steps and (2) dynamic window selection, where windows are dynamically chosen by evaluating policies at each noise step. In the dynamic approach, we identify the step $t$ that optimizes the sampling trajectory by finding the policy $\pi(a_t \mid s_t)$ that *maximizes reward variation while minimizing divergence from the image prior* $\rho(x_{\mathrm{noised}})$. This reward is given by:

$$R_{\theta,t} = \text{Reward from } \tilde{x}_{t \to 0} \mid \pi(a_t \mid s_t), \quad \pi_{\text{best}}(a_t \mid s_t) = \max_t (R_{\theta,t} - R_{\theta,t+1}) \ \forall t \in [T-1, 0].$$

where $\tilde{x}_{t \to 0}$ is the noise-free version of $x_t$. We incorporate a diversity-promoting term by considering the mean distance between samples (at each step) and the prior. The optimal policy is defined as:

$$\pi_{\text{best}}(a_t \mid s_t) = \max_t \left( (R_{\theta,t} - R_{\theta,t+1}) - \beta \cdot D(\rho_0(s_0), \tilde{x}_{t \to 0}) \right) \quad \forall t \in [T-1, 0],$$

where $\beta$ is a hyperparameter and $D$ is a distance metric such as cosine distance:

$$D(\rho_0(s_0), \tilde{x}_{t \to 0})) = \text{mean} \left( D_{\cos}(\rho_0(s_0), \tilde{x}_{t \to 0})) \right) \quad \forall \text{ samples at step } t.$$

With this, our training process is as follows:

*Algorithm 1: Initial Step.* For a predefined window, a sampling cluster $C$ is generated for each epoch. For each sample within the batch, a starting step $t_i$ is sampled from a uniform distribution over the cluster $C$, and the final state $s_0$ for the current batch is generated. If we are dynamically selecting the initial step, we sample all possible intermediate states and select the ones that maximizes the objective detailed above in section 4.3.

*Algorithm 2: Hierarchical Training.* It extends this sampling strategy to the full batch training process. Initially, several reference images are sampled from the selected initial steps (Initial Step). For each batch, noise is added to the final state of the reference image per batch $s_0$ to create new initial states $\{s_i\}_{i=t}^{\text{num\_batches}}$. The algorithm then generates trajectories for a specified number of samples from the corresponding starting steps and applies DDPO to each clipped trajectory.

---

**Algorithm 1** Initial Step

1: **for** epoch $e$ **do**
2:    **if** Predefined Window **then**
3:       Sample steps $t_i \sim \mathcal{U}(C)$ and generate final state $s_0$.
4:    **else if** Dynamic Window Selection **then**
5:       **for** sample $i$ in batch size $b$ **do**
6:          Add noise, denoise $\{s_t\}$, compute rewards $R_{\theta,t}$, and select step $\pi_{\text{best}}(a_t \mid s_t)$.
7:       **end for**
8:    **end if**
9: **end for**
10: **return** Final states and steps $(\{s_0\}_{i=t}^b, t_i)$.

---

**Algorithm 2** Hierarchical Training

1: **Initial state:** Sample num_batches with Initial Step
2: **for** batch $j$ in num_batches **do**
3:    Add noise to $s_0[\text{num\_batches}]$ up to the new initial state $\{s_i\}_{i=1}^{\text{num\_batches}}$
4:    Get trajectory for $n_{\text{samples}}$ samples.
5:    Apply DDPO hierarchical for trajectory $[s_i : \mathrm{x}_0]$
6: **end for**

---

## 5 EXPERIMENTS

### 5.1 TASKS AND REWARD FUNCTIONS

We employed three downstream tasks as outlined in Black et al. (2024): compressibility, incompressibility, and aesthetic quality. The first two tasks are defined by the size of images after applying a JPEG compression algorithm, serving as the reward function. For aesthetic quality, we utilized the LAION aesthetic model (Schuhmann, 2022), a multilayer perceptron that assigns a scalar value from 1 to 10 to indicate the aesthetic quality of an image.

These tasks show RL's ability to optimize objectives like compressibility, which are hard to encode in a loss function. The LAION aesthetic model further demonstrates how RL leverages human feedback to align diffusion models (Ouyang et al., 2022).

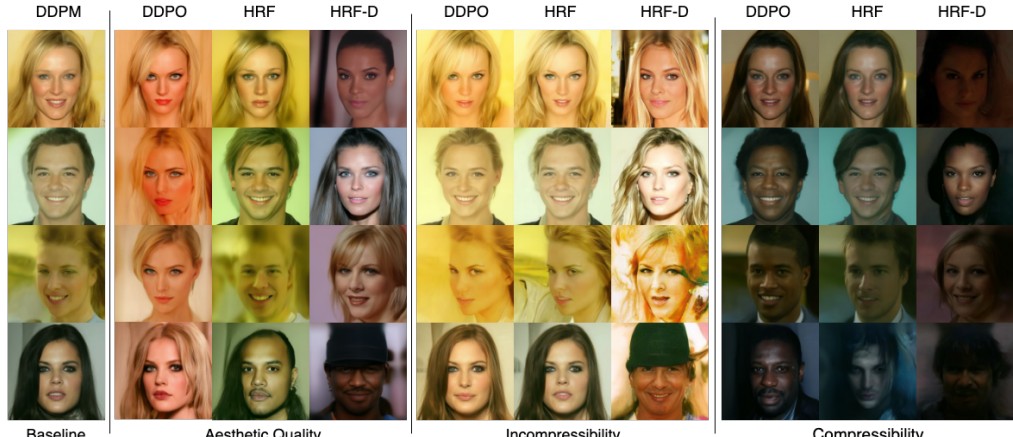

Figure 4: **Alignment of diffusion model to downstream tasks**. This figure shows the visual performance of a diffusion model on three tasks: aesthetic quality, incompressibility, and compressibility. **DDPO** and **HRF** achieve similar semantic changes, but **HRF** better preserves visual diversity, especially for aesthetic quality. While **DDPO** risks mode collapse by generating similar high-reward images, **HRF** maintains diversity while improving rewards. **HRF-D** shows a significant visual shift, with samples differing greatly from the originals but maintaining high diversity. Compressibility skews towards darker samples, yet retains diverse representations.

## 5.2 EXPERIMENTAL SETUP

We train our base model using DDPO and our proposed DDPO-based methods with three random seeds each, applying early stopping at a common stable point. First, we train DDPO to a target reward, then fine-tune our hierarchical models to achieve similar scores, resulting in 16 models: one baseline, one DDPO per task, three HRF per task, and one HRF-D per task. We evaluated these models via the following performance indices:

1. **Inception Score (IS) Salimans et al. (2016)**: Measures both sample quality and diversity by evaluating the probability of generated samples belonging to distinct classes. A higher score indicates better visual quality and diversity, while a score closer to baseline score reflects preserved sample diversity and distribution.
2. **Vendi Score** (Dan Friedman and Dieng, 2023), which measures diversity as the exponential of the Shannon entropy of a similarity matrix. It approximates the effective number of distinct classes in a sample based on a distance metric. For our evaluation, scores closer to baseline indicate better sample diversity preservation.
3. **Reward Score**: Evaluates a model's adaptability to a given downstream task by assigning a numerical value to task-specific performance. Higher scores indicate better alignment with the task, providing an intuitive measure for comparing model behavior under similar conditions.

Hyperparameter settings and resources can be found in Appendix A.2.

## 5.3 ABLATIONS ON WINDOW SELECTION IN HIERARCHICAL REWARD FINE-TUNING

We performed a sensitivity analysis on window selection based on three regimes: baseline, early, and later stages, each trained with three seeds. With 40 sampling steps (*0 = noisiest, 40 = final image*), the windows used are:

1. **Baseline**: [(8,12), (18,22), (28,32)]—three equidistant windows.
2. **Early**: [(3,7), (18,22), (28,32)]—first window shifted to noisier states.
3. **Later**: [(8,12), (28,32)]—middle window removed, iterating more on the last window.

Figure 5 shows that different rewards favor different strategies: *Aesthetic Quality* prefers less noisy windows, while *Compressibility* and *Incompressibility* favor noisier ones.

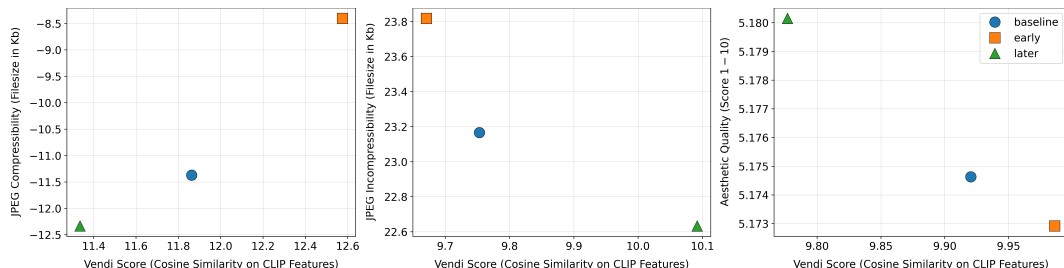

Figure 5: **Rewards (y-axis) vs Diversity (x-axis, Vendi Score)**. Ablations on window selections for three hierarchical regimes: baseline (blue), early (orange) and latter stages (green) over the three downstream tasks considered. Each result is reported as the average of three-run seeds.

## 5.4 RESULTS

Table 1: **Reward Mean and standard error for each downstream task across using `google/ddpm-celebahq-256` as pretrained model**. All samples were generated using the same initial noise to ensure a fair comparison. **Baseline** refers to the generative capabilities of the pretrained model. **DDPO** displays results from the fine-tuned models using DDPO with importance sampling (see Section 3.2). **HRF** represents our proposed method based on the average results reported on different window selection schemas (see Section 5.3). On the other side, **HRF-D** results are obtained using a *dynamic window selection* method.

| Downstream Tasks | Baseline | DDPO | HRF | HRF-D |
|---|---|---|---|---|
| Aesthetic Quality (↑ better) | $5.11 \pm 0.01$ | $\mathbf{5.55} \pm 0.01$ | $5.18 \pm 0.01$ | $5.41 \pm 0.01$ |
| Compressibility (↓ better) | $17.26 \pm 0.05$ | $5.30 \pm 0.06$ | $8.40 \pm 0.08$ | $\mathbf{5.20} \pm 0.07$ |
| Incompressibility (↑ better) | $17.26 \pm 0.05$ | $21.59 \pm 0.08$ | $23.81 \pm 0.10$ | $\mathbf{38.77} \pm 0.15$ |

Table 2: **Sample Diversity Assessment Using Inception Score.** The Inception Score (IS) represents the mean value calculated across $2,142$ images, with values closer to baseline indicating better diversity preservation.

| Downstream Tasks | Baseline | DDPO | HRF | HRF-D |
|---|---|---|---|---|
| Aesthetic Quality | $2.07 \pm 0.03$ | $1.58 \pm 0.02$ | $\mathbf{2.04} \pm 0.03$ | $1.91 \pm 0.03$ |
| Compressibility | $2.07 \pm 0.03$ | $2.28 \pm 0.03$ | $\mathbf{2.16} \pm 0.03$ | $2.26 \pm 0.03$ |
| Incompressibility | $2.07 \pm 0.03$ | $1.99 \pm 0.05$ | $\mathbf{2.10} \pm 0.04$ | $2.39 \pm 0.05$ |

We evaluated the baseline DDPM model's generation capacity, measuring the mean reward over 2,142 samples for the three downstream tasks considered. Reproducing the DDPO method from Black et al. (2024), our approach outperformed the baseline across all tasks considered (see Table 1).

The HRF and HRF-D methods performed on par with –or better than– standard DDPO. While achieving similar mean rewards, our approach successfully optimized the reward without collapsing samples into a single mode, thus preserving diversity. This claim was validated via the Inception Score (IS) (Salimans et al., 2016; Heusel et al., 2017) as described in Section 5.2. Table 2 shows that HRF and HRF-D achieved IS values closer to the baseline, indicating improved sample diversity over DDPO while successfully fine-tuning the model for the tasks of interest.

Table 3 further validates our claims. Vendi Scores closer to the baseline indicate better diversity preservation, and both HRF and HRF-D outperformed DDPO in that regard. While higher Vendi scores typically mean more diversity, our goal is to maintain diversity within the support of the original distribution, making excessive diversity undesirable for this study. These metrics were computed using $2,142$ samples. For more details on how the scores change as the number of samples increases, refer to Appendix A.4.

Table 3: **Sample Diversity Assessment Using Vendi Score computed using cosine similarity metric on raw pixels and CLIP embeddings.** Vendi score measures the number of distinct classes: scores closer to the baseline preserve the same number of unique samples, indicating better sample diversity preservation. Higher values show greater diversity but do not guarantee preservation within the original distribution support.

| Vendi Score | Raw Pixels | | | | CLIP Embeddings | | | |
|---|---|---|---|---|---|---|---|---|
| Downstream Tasks | DDPM | DDPO | HRF | HRF-D | DDPM | DDPO | HRF | HRF-D |
| Aesthetic Quality | 2.80 | 1.53 | **2.74** | **2.46** | 10.34 | 4.55 | **9.89** | **9.59** |
| Compressibility | 2.80 | 6.51 | **3.90** | **3.53** | 10.34 | 13.04 | **11.92** | **12.80** |
| Incompressibility | 2.80 | 2.05 | **2.48** | **3.20** | 10.34 | 8.63 | **9.84** | **10.30** |

The evidence shows a trade-off between optimizing for a downstream objective and maintaining sample diversity, as reinforcement learning inherently skews samples toward policy preferences. Our approach appears to balance this trade-off by focusing on learning dynamics and training in a more controlled step-wise manner, rather than manipulating the reward.

In summary, our findings suggest that our RL-based training approach achieved comparable performance to DDPO in reward optimization, while better preserving model diversity. This highlights the robustness and efficacy of HRF and HRF-D as a method, showcasing its potential for enhancing model performance across various tasks while avoiding mode collapse.

## 6 CONCLUSIONS

We have presented HRF, a novel methodological framework that leverages the hierarchical nature of data generation in diffusion models and improves the multi-step decision-making interpretation. In this line, we have also introduced a phase training scheme that enhances specific parts of the diffusion model during fine-tuning. The method provides new insights into learning via reinforcement learning and the effects on each diffusion step. Our hierarchical formulation has proven to be an effective method for fine-tuning diffusion models on downstream tasks while successfully preserving the inherent models' diversity. Our evaluation utilized known and validated metrics such as the Inception Score (IS) and Vendi Score to measure the performance of the generative models considered, thus providing a trustworthy assessment of the

**Future work.** Some future work directions include optimizing cluster selection and hyperparameters to balance learning and sample diversity. Also, exploring alternative RL formulations within this framework could yield interesting results. While we showed that manual step selection improves performance, finding optimal training scenarios remains an open question that we initially explored with HRF-D, which employs an algorithm to identify optimal training steps, but it is not yet fully optimized and requires further research. Finally, we find it extremely necessary to define benchmarks with specific pre-trained models and tasks to measure and compare different post-training strategies, as this is an area rather unexplored. Methodologies like the one proposed in this paper would benefit from better and more concise evaluation.

**Limitations.** Our method aims to control diffusion models to align with downstream tasks via reward functions, offering benefits like avoiding mode collapse. However, risks include generating explicit content, and more research is needed to understand its broader impact.

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

# A  APPENDIX

## A.1  INJECTION SAMPLING

When investigating the impact of learning with the MDP formulation we propose that most of the learning prowess is being done at the early steps of the diffusion model (the noisier states). To verify this experimentally we propose the following method:

1. Sample 15 trajectories with all fine-tuned models using DDPO on three tasks.
2. Select initial injection steps: $[38, 35, 30, 25, 20, 10]$, where 40 is the noisiest and 0 is the final sample.
3. Resample 15 trajectories, switching to the base model at selected injection steps, and compute all trajectory variations.
4. Compute cosine distances between denoised images of the original and fine-tuned models for all starting timesteps for the same starting seed. This is between the estimated noiseless image for each timestep.
5. Plot the distances for all three models.

Following these steps we get the differences that each model makes for the denoising trajectory at different timesteps. We confirm that most of the differences in the final image come from the early stages of the diffusion model 6. This insight further solidifies the fact that almost all of the learning tends to happen at the early stages of the diffusion model due to the training inherent dynamics.

## A.2  IMPLEMENTATION DETAILS

We provide the implementation details we used, such as computational resources and relevant hyperparameters settings. All this hyperparameters may be subject to changes and exploration, especially in the HRF implementations. Where the hyperparameters where found experimentally.

### A.2.1  RESOURCE DETAILS

**For HRF**: GPU experiments were conducted on the default NVIDIA system within the cloud computing provider Hyperstack, with one A100 Tensor Core GPU and 150GB of GPU memory. The training time for each iteration took approximately 4 hours. Inference for the reward model was performed on a single NVIDIA L4 GPU and takes about 2 minutes per 40 images.

## A.3  FULL HYPERPARAMETERS

## A.4  VENDI SCORE

We conducted an empirical analysis of the incrementally computed Vendi Score using the cosine distance metric on CLIP embeddings. The results are shown in Figure 7 . The analysis begins with an initial sample size of $500$, increasing by 5 samples at each step until reaching $2000$. Afterwards, the sample size is increased by $500$ samples per step until reaching a total of $11,000$ samples. We ended up using $2,142$ samples to compute the Vendi Score, capturing the major increase in diversity.

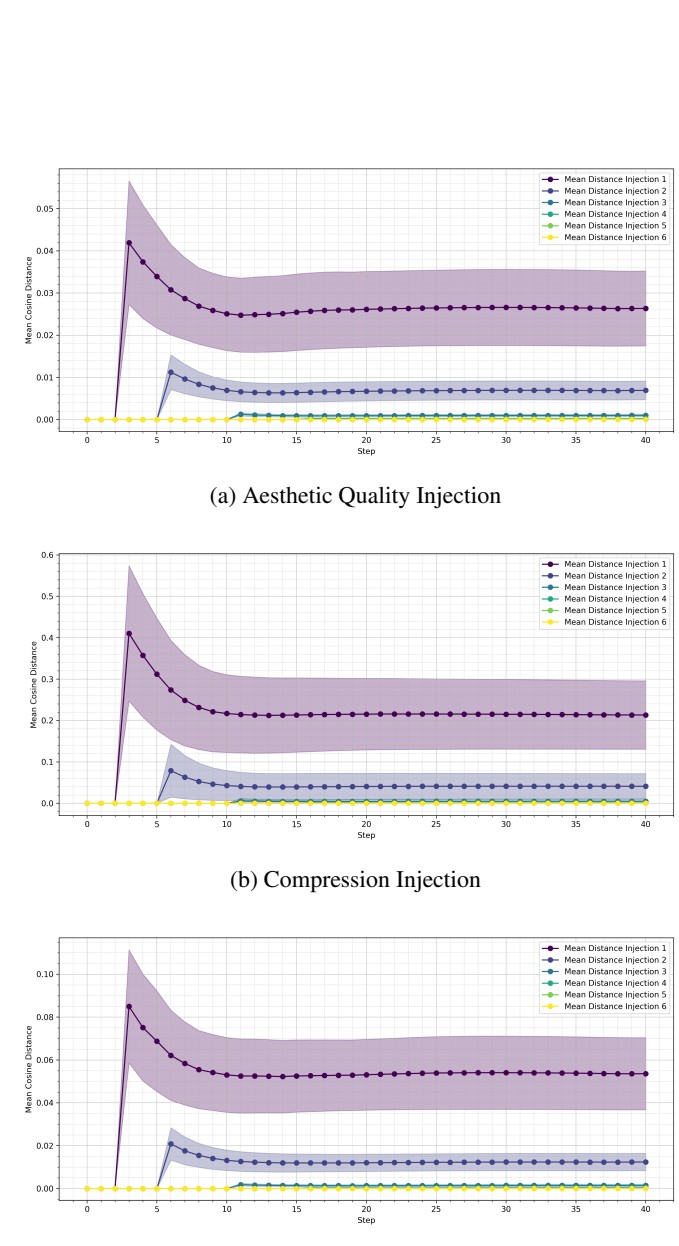

(a) Aesthetic Quality Injection

(b) Compression Injection

(c) Incompression Injection

Figure 6: Cosine Distance for all three injection experiments. We see in all of them a strong tendency to learn the definitive features early in the diffusion chain. And after injection sampling with the baseline model, we see little variation even at the midway point.

Table 4: Full Hyperparameters for Aesthetic Quality

| | DDPO$_{IS}$ | HRF | HRF-D |
|---|---|---|---|
| **Diffusion** | | | |
| Denoising steps ($T$) | 40 | 40 | 40 |
| **Optimization** | | | |
| Optimizer | AdamW | AdamW | AdamW |
| Learning rate | 3e-7 (With warm up) | 3e-7 | 9e-6 |
| Weight decay | 1e-3 | 1e-3 | 1e-3 |
| Gradient clip norm | 4.5 | 4.5 | 4.5 |
| **HRF and HRF-D** | | | |
| Batch size | - | 4 | 25 |
| Samples per iteration | - | 160 | 150 |
| Gradient updates per iteration | - | 1 | 1 |
| Clip range | - | 1e-4 | 1e-4 |
| Clusters | - | [(3,7),(18,22),(28,32)] | - |
| Number of iters per cluster | - | [8,8,8] | - |
| beta | - | - | 1.0 |
| **DDPO** | | | |
| Batch size | 10 | - | - |
| Samples per iteration | 100 | - | - |
| Gradient updates per iteration | 1 | - | - |
| Clip range | 1e-4 | - | - |

Table 5: Full Hyperparameters for Compressibility and Incompressibility

| | DDPO$_{IS}$ | HRF | HRF-D |
|---|---|---|---|
| **Diffusion** | | | |
| Denoising steps ($T$) | 40 | 40 | 40 |
| **Optimization** | | | |
| Optimizer | AdamW | AdamW | AdamW |
| Learning rate | 9e-7 | 9e-7 | 9e-7 |
| Weight decay | 1e-3 | 1e-3 | 1e-3 |
| Gradient clip norm | 4.5 | 4.5 | 4.5 |
| **HRF and HRF-D** | | | |
| Batch size | - | 10 | 25 |
| Samples per iteration | - | 150 | 150 |
| Gradient updates per iteration | - | 1 | 1 |
| Clip range | - | 1e-4 | 1e-4 |
| Clusters | - | [(8,12),(18,24),(28,36)] | - |
| Number of iters per cluster | - | [8,8,8] | - |
| Beta | - | - | 1.0 |
| **DDPO** | | | |
| Batch size | 10 | - | - |
| Samples per iteration | 150 | - | - |
| Gradient updates per iteration | 1 | - | - |
| Clip range | 1e-4 | - | - |

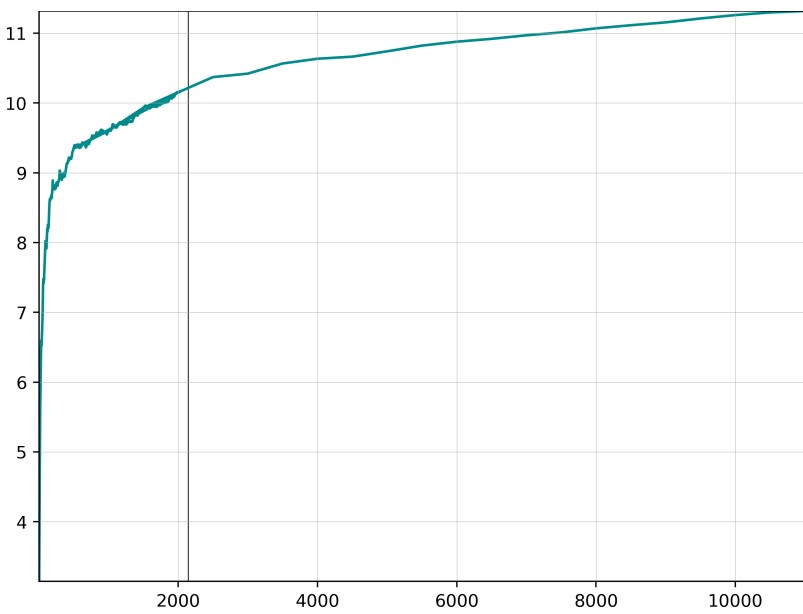

Figure 7: **Incremental Vendi Score based on cosine distance over CLIP embeddings and the number of DDPM samples (x-axis).**. The analysis starts with 500 images, increasing by 5 samples per step until reaching 2,000, after which steps increase by 500 samples up to a total of 11,000. The vertical line marks the Vendi Score at 2,142 samples, the same number used for diversity evaluation in ablation studies and Table 3 This consistent sample size ensures a fair comparison of diversity across different experimental conditions.

