# OpenReview forum: "Avoiding mode collapse in diffusion models fine-tuned with reinforcement learning"
_ICLR.cc/2025/Conference — Submitted to ICLR 2025_

### Official Review · Reviewer_7Vfa · 2024-10-20

**Soundness:** 2
**Presentation:** 2
**Contribution:** 2
**Rating:** 3
**Confidence:** 3

**Summary:**

This paper introduces an approach called Hierarchical Reward Fine-tuning (HRF), which aims to address training instability and mode collapse issues when fine-tuning diffusion models with reinforcement learning (RL), specifically by improving the diversity of generated samples. The algorithm builds upon the Denoising Diffusion Policy Optimization (DDPO) framework and incorporates two window selection schemes to enhance variation between consecutive steps while minimizing divergence from the prior. Experimental results are provided to demonstrate the effectiveness of the proposed method.

**Strengths:**

* The problem is motivated well.

* Related work and background are adequately discussed.

**Weaknesses:**

* This approach does not differ significantly from the original DDPO method. For instance, the content in Sections 3.1-3.2 and 4.2 is very similar to Sections 3-4 of the DDPO paper. It seems that the core contribution of this work lies in Section 4.3, where the window selection schemes are proposed. These schemes should be discussed in greater detail, as they appear to be the main novel aspect of this paper.

* Section 4.3 is poorly written compared to the earlier sections. For example, the definition of $\pi_{best}$ is presented twice, which could lead to ambiguity. Additionally, the algorithm tables are too informal, making implementation difficult. It would also be helpful to introduce the definition of the cosine distance for the distance metric $D$, as this would provide greater clarity.

* In Tables 4 and 5, it appears that this paper uses different hyperparameters for DDPO compared to the original settings in the DDPO paper, and this discrepancy should be explained. Additionally, in deep RL, the number of gradient updates per iteration is closely related to the number of samples and the batch size. Typically, PPO performs multiple gradient updates at each iteration. Therefore, having only one gradient update per iteration across all algorithms seems very unusual.

**Questions:**

* How is the objective function (5) optimized? Does it use proximal policy optimization (PPO) as in the DDPO paper or trust region policy optimization (TRPO)? The only information I could find is "We also employ trust regions to address inaccurate estimations of $p_\theta$." in line 296-297.

* What is the architecture of the model $\pi_\theta$?

* Can we use metrics other than cosine distance to measure the distance between $\rho_0(s_0)$ and $\tilde{x}_{t \rightarrow 0}$?

---

### Official Review · Reviewer_NqvS · 2024-10-25

**Soundness:** 3
**Presentation:** 3
**Contribution:** 2
**Rating:** 5
**Confidence:** 3

**Summary:**

Previous methods suffer from issues such as training instability and mode collapse. To address these challenges, the authors propose HRF, which utilizes a window selection strategy, avoiding the full denoising trajectory by starting at intermediate time steps. They introduce two strategies: manual and dynamic window selection. From these intermediate steps, HRF explores diverse trajectories by adding noise, and it is trained similarly to the previous DDPO approach.

**Strengths:**

1. The authors proposed a new fine-tuning method which is an extension of DDPO to address the collapse issues in the diffusion model.
2. The authors provided clear motivation, e.g., Figure 1
3. They showed well-structured experiments to validate their proposed method HRF.

**Weaknesses:**

1. Figure 4 is unclear to support the effectiveness of HRF compared to the previous methods
2. No explicit reasons between the performances between the two strategies (manual vs dynamic)

**Questions:**

**Q1:** In my view, the Figure 4 is unclear to show the outperforming performance of HRF compared to other baselines. I wonder if the reviewers provide much more clearer figures to show the effectiveness of HRF

**Q2:** In Table 4 and Table 5, (HRF and HRF-D) and (DDPO) are separated although all parameters of DDPO belong to those of  (HRF and HRF-D). It would be better if the authors combined them. In addition, I recommand that the reviewers check typos in the supplementary materials.

**Q3:**  HRF uses the predefined windows, and the authors used the uniform distribution over the cluster. If the authors can provide reasonable pre-defined non-uniform distributions with some experimetal results, I think that it must be useful to improve the quality of this paper. I would like to hear the authors' opinions.

**Q4:** Comparing HRF and HRF-D, it looks like that the performance gap between them is marginal. Thus, one may think that sliding strategy seems to be marginal. Therefore, I believe that an ablation study without the sliding strategy, using DDPO with noisy initial states to generate diverse trajectories, could enhance the quality of this paper.

**Q5:** It would be beneficial if the code could be made publicly available in the future. Some parameters of interest are not clear. For instance, the definition of epoch $e$ for each task is being unclear in the paper from my perspective.

---

### Official Review · Reviewer_CKPK · 2024-10-27

**Soundness:** 3
**Presentation:** 2
**Contribution:** 2
**Rating:** 5
**Confidence:** 3

**Summary:**

This paper introduces Hierarchical Reward Fine-tuning (HRF), a novel approach to address mode collapse when fine-tuning diffusion models with reinforcement learning. It proposes two sliding-window approaches to sample intermediate diffusion steps as initial starting points for the fine-tuning. The experiments validate the contribution of the paper, showing an increase in output diversity.

**Strengths:**

1. The paper addresses an important and relevant problem when optimizing/fine-tuning diffusion models with RL.
2. The proposed method is sound and novel to me.

**Weaknesses:**

1. The proposed window selection mechanism seems handcrafted and requires a lot of tuning. This may limit the algorithms' further application. For example, when the diffusion step increases, you have to manually define the windows.
2. Have you considered randomly sampling the noise step $t$? how is it compared to the two proposed mechanism?
3. The proposed window selection mechanism, especially the HRF-D, introduces a lot of additional computation.
3. It seems unfair to fine-tune based on DDPO. It is better to either fine-tuning the pre-trained model with HRF or extending DDPO training to equalize training steps.
4. The writing can be improved a lot. For example, the symbol $i$ is inconsistently defined throughout. In addition, it is a big claim to say "avoid" in the title. I suggest the author to moderate the claim a bit.

**Questions:**

1. What is $c$ in Eq. 5?
2. Can you elaborate more on why different windows have different impacts on compressibility and aesthetic quality?

---

### Official Review · Reviewer_U5SJ · 2024-11-03

**Soundness:** 2
**Presentation:** 2
**Contribution:** 1
**Rating:** 3
**Confidence:** 3

**Summary:**

This paper proposes a new fine-tuning method for diffusion foundation models based on reinforcement learning. With the hierarchical interpretation of diffusion models, the proposed method, called Hierarchical Reward Fine-tuning can better preserve the sample diversity in DDPM compared to the previous DDPO method.

**Strengths:**

The paper is overall well-written and the idea of hierarchically-finetuning the diffusion foundation model is interesting.

**Weaknesses:**

1.	The proposed method seems a little bit trivial. In the reviewer’s opinion, it just chooses a specific diffusion step and applies RL-based finetuning after this step.
2.	The hierarchical nature seems limited to images. It is hard to say this phenomenon still holds in other applications of diffusion model such as text, audio and decision-making.
3.	The authors need to conduct more experiments on different datasets/diffusion foundation models and give more example images. Otherwise, the results is not convincing to confirm the efficiency of their methods.
4.	In line 200, x_0 should be set before the period or full stop.
5.	Some usage of notation confuses me in this paper. For instance, in line 326, why $\pi_{best}(a|s)$ is the difference of reward rather than probability density? In line 332, $\tilde{x}$ is an estimated image but $\rho_0$ is a combination of distribution. How can we measure the distance?

**Questions:**

1.	What does the sampling cluster mean in line 340?
2.	Can the author show more examples and results on more datasets to show the superiority of their method? I am willing to raise my score with more convincing results.

---

### Meta-Review · Area_Chair_gaJv · 2024-12-21

**Metareview:**

This paper addresses the issue of model collapse in fine-tuning diffusion models. The proposed approach utilizes the hierarchical structure of the diffusion model, uses a window selection strategy, avoids full denoising, and is shown to have more diversity in generation, indicating avoidance of collapse.

The algorithm contains many components, requiring careful ablation studies that are not present. Specifically, the handcrafted nature and computational cost of the window selection mechanism require more careful design and study of the algorithm. The focus on image-only tasks is also limiting.

**Additional Comments On Reviewer Discussion:**

The reviewers unanimously felt the paper was not ready. The reviewers found the method too trivial, not well-explained (unclear presentation) or limiting (specific to images). The comparisons with DDPO are seen by the reviewers as biased due to using different setups and hyperparameter choices​ than the original. The authors did not respond to reviewers’ criticisms.

---

### Decision · Program_Chairs · 2025-01-22

Reject